# OpenReview forum: "Mixture of insighTful Experts (MoTE): The Synergy of Thought Chains and Expert Mixtures in Safety Self-Alignment"
_ICLR.cc/2025/Conference — Submitted to ICLR 2025_

### Official Review · Reviewer_TQAf · 2024-11-01

**Soundness:** 2
**Presentation:** 2
**Contribution:** 2
**Rating:** 5
**Confidence:** 2

**Summary:**

This paper proposes a self-alignment method that utilizes a Chain of Thought approach (AlignCoT) to address two challenges: 1. Recent studies rely on the assumed emergent capabilities of LLMs.  2. Recent studies discard all intermediate reasoning steps, when aligning the model with updated answers.

**Strengths:**

This paper shows the promising performance of AlignCoT and MoTE based on Alpaca-7B and Wizard-Vicuna-Uncensored 7B.

**Weaknesses:**

1. In Table 1, it is unclear if the results are based on Alpaca-7B or Wizard-Vicuna-Uncensored 7B.
2. For evaluation, it would make this paper stronger if the authors could provide more details about how human annotators provide a final verification for precise results. For example, what are the annotation guidelines provided to annotators, how disagreements between annotators were resolved, how human annotators are picked, and how many human annotators are in the verification process?
3. It would be better if the authors could show if their method has the same effects in bigger LLMs (such as picking some LLMs with 13B or even larger).
4. The metrics, help, harm, and harm-help are too vague and may be sensitive to human annotators. It would make this paper stronger if the authors could provide better metrics. For example, how the authors deal with the annotator bias or subjectivity in current metrics.

**Questions:**

Check the weaknesses section above.

---

> ### Author Response · Authors · 2024-11-25
> **Response to Reviewer TQAf**
>
> Thank you for your detailed feedback, which has helped strengthen our work. We have addressed each point you raised with specific clarifications and improvements.
>
> **1. Table 1, the results are based on which model?**
>
> For PKU-SafeRLHF, the results are based on Alpaca-7B, and for HH-RLHF, they are based on Wizard-Vicuna-Uncensored-7B.
>
> **2. For evaluation, it would make this paper stronger if the authors could provide more details about how human annotators provide a final verification for precise results. \& The metrics, help, harm, and harm-help are too vague and may be sensitive to human annotators.**
>
> We use GPT-4 to evaluate the helpfulness and harmfulness of responses rather than relying on human annotators. This approach aligns with most baselines, and the relationship between GPT scores and human judgments has been extensively analyzed in [1, 2]. Based on their findings, we directly adopt these metrics, ensuring consistency with established practices.
>
> [1] Judging LLM-as-a-Judge with MT-Bench and Chatbot Arena, NeurIPS 2023
>
> [2] RLCD: Reinforcement Learning from Contrastive Distillation for Language Model Alignment, ICLR 2024
>
> **3. It would be better if the authors could show if their method has the same effects in bigger LLMs (such as picking some LLMs with 13B or even larger).**
>
> We applied MoTE to the larger Llama2-13B model and plan to explore even larger models in future work. As shown in the following table, MoTE remains effective on the 13B model and achieves new state-of-the-art results on HH-RLHF.
>
> |                       | **Help** | **Harm** | **Harm-Help** |
> |---------------------------|----------|----------|---------------|
> | **current SoTA (MATRIX)** | 6.93     | 87.55    | 5.46          |
> | **llama2-13b**            | 7.20     | 91.39    | 6.55          |
> | **MoTE-llama2-13b**       | 7.21     | 94.73    | 6.93          |

---

> > ### Comment · Reviewer_TQAf · 2024-11-30
> >
> > Thank you for your response. After carefully considering your rebuttal and the concerns raised by other reviewers, I have decided to maintain my original score.

---

### Official Review · Reviewer_aK5g · 2024-11-02

**Soundness:** 4
**Presentation:** 4
**Contribution:** 3
**Rating:** 6
**Confidence:** 3

**Summary:**

The paper introduces AlignCoT a self-alignment method for generating dataset that facilitating the safe responses of LLMs. Furthermore, the paper also proposes Mixture of Insightful Expert (MoTE) that extends a mixture of experts (MoE) framework with a share expert to facilitate the exchange of knowledge across different stages of AlignCoT. The two contributions are relatively orthogonal, but the sufficient experiments are conducted to demonstrate the necessity of both methods on HH-RLHF benchmark.

**Strengths:**

1. The paper demonstrates comprehensive experiments, draws comparison with enough baselines, and carefully examines multiple design choices in both AlignCoT and MoTE.
2. The paper is very-well organized and the presentation is clear.

**Weaknesses:**

1. The benefit of the shared expert in MoTE is a bit unclear. It sounds like a mixture of MoE with single expert LoRA. The authors mentioned information exchange. This is very generic, does MoTE extends to other self-alignment methods?
2. The novelty from each method is relatively limited. For example, AlignCoT is just one of CoT frameworks. A large amount of CoT frameworks have been explored these days. Also MoTE sounds like a mixture of MoE with single expert LoRA. Although, tricks like step skipping and the process to streamline the training of MoTE have been explored. The overall novelty of MoTE compared to MoE is relatively limited.

**Questions:**

1. Do author expect MoTE to be extended to other self-alignment methods that involves multiple experts? Could the authors provide more insight on why shared expert can improve the performance?
2. I am a bit confused by single-step inference (line 161) and figure 2. Why the paper aims for improving the single-step inference? Single_Sequence dataset is crafted for Single-step Inference setting, but MoTE can even achieve better performance. Could the author elaborate more on this result?

---

> ### Author Response · Authors · 2024-11-25
> **Response to Reviewer aK5g**
>
> Thank you for your detailed feedback, which has helped strengthen our work. We have addressed each point you raised with specific clarifications and improvements.
>
> **1. The benefit of shared expert.**
>
> The shared expert is a critical design element that enables the answer expert $E_{ans}$ to internalize information from analysis and guidance, making it essential for single-step inference. Without the shared expert, $E_{ans}$ cannot interact effectively with $E_{a}$ (analysis expert) and $E_{g}$ (guidance expert).
>
> For instance, under single-step inference, activating only $E_{ans}$ without the shared expert reduces MoTE to a naive safety alignment approach, relying solely on QA pairs without leveraging training with analysis and guidance. As shown in the ablative analysis in Table 3, the safeness of MoTE without the Shared Expert drops significantly. This demonstrates that the shared expert is a key design for fostering synergy among experts and enhancing the effectiveness of thought chains.
>
> **2. Can MoTE be extended to other self-alignment methods that involves multiple experts?**
>
> Yes, MoTE can be extended to other self-alignment methods that utilize multiple experts. In this work, we use AlignCoT as an example to analyze the integration of Mixture of Experts (MoE) and Chain-of-Thought (CoT) in safety alignment. While the current implementation focuses on this framework, MoTE can be straightforwardly adapted to other alignment methods involving additional experts, which we plan to explore in future work.
>
> **3. Novelty of each component is relatively limited**
>
> We claim that we are the first to integrate Mixture of Experts (MoE) into safety self-alignment and have designed an efficient framework to achieve this. We have highlighted the importance of each component and analyzed the overall efficiency of our framework. We believe this work will contribute to the community by facilitating the extension of MoE into alignment methodologies.
>
> **4. Why aim for single-step inference and why MoTE is effective on this setting?**
>
> Single-step inference is critical in the safety domain, as most harmful questions are immediately recognizable to humans at first glance. Thus, our goal is to make safety an instinctive response for the LLM, avoiding the need for generating lengthy and tedious thought chains. Notably, all baseline methods are also designed with single-step inference in mind, ensuring fair comparison.
>
> MoTE performs better in this setting due to two key reasons. Theoretically, as explained in Appendix A (now moved to the main text, Lines 160-183), learning intermediate steps collectively enhances single-step inference performance. Practically, MoTE trains specialized experts for different tasks—analysis, guidance, and answering—to strengthen each intermediate step (Fig. 2). Additionally, step-skipping further improves single-step performance (Table 3). These factors demonstrate that MoTE is particularly effective for safety self-alignment in single-step inference scenarios.

---

### Official Review · Reviewer_S7t5 · 2024-11-03

**Soundness:** 2
**Presentation:** 1
**Contribution:** 2
**Rating:** 3
**Confidence:** 3

**Summary:**

This paper presents a method to improve the safety alignment of LLMs using AlignCoT, a Chain-of-Thought (CoT) based prompting technique that structures datasets with intermediate reasoning steps, and MoTE, a multi-LoRA approach where different parts of the sequence embedding are processed by specialized LoRA modules. The authors train and evaluate their approach on the Safe-RLHF and HH-RLHF datasets, assessing both helpfulness and harmlessness in single-step and multi-step inference settings.

**Strengths:**

- The paper provides detailed ablation studies, clearly analyzing the contributions of each component, including AlignCoT and MoTE, to the overall performance.

**Weaknesses:**

- The formulations in Section 3, including Equation 1, are incomplete and should be revised with more standard notations. For instance, rather than using 𝑥 for context-response pairs, (𝑥,𝑦) should be used, and the source (policy/distribution) of 𝑥 should be clarified. Additionally, the justification for modeling CoT as a joint probability distribution between different steps is not clear. If this formulation is kept, parts of Appendix A should be integrated into the main text for clarity.

- The related work section lacks coverage of self-alignment methods and other fine-tuning approaches based on CoT. A dedicated subsection comparing AlignCoT with previous work is needed to clarify its novelty.

- Despite the point above, Section 3 provides minimal technical discussion. The formulation could be more effectively replaced with a figure illustrating the structural prompting process.

- The "Efficient training of MoTE" section offers little novelty, as its function appears similar to existing masked attention techniques.

- The paper's clarity needs improvement. For example:
  - It is unclear which 7B LLM is referred to in Line 177. Is it Alpaca-7B?
  - The metrics used in Figure 2 are not specified or explained in the text.

- Important details about the methods used in training and evaluation should be moved from Appendix B to the main text.

- The paper's reliance on Alpaca-7B as a baseline is concerning, especially given the rapid advancements in LLM alignment and training. Using an older model might mean addressing issues that have already been mitigated in more recent models. This raises the question of whether the improvements demonstrated are still relevant in the context of modern LLMs. It would be more convincing to evaluate the method on current models like Llama-3-8B, Llama-2-7B, or Mistral-7B, which better reflect the state-of-the-art in language model performance and alignment challenges.

- The evaluation presented in Table 1 is somewhat narrow, relying heavily on PKU-SafeRLHF and HH-RLHF, which are closely related datasets, which limits the evaluation's breadth. Cross-testing results should be included, and other benchmarks like XSTest, WildChat, or win rate comparisons against Llama-Guard or Perspective API (less recommended) should be considered.

- Unless I overlooked it, the code for the proposed methods has not been provided. This makes it difficult to verify the results or apply the techniques described in the paper.

**Questions:**

See Weakness.

---

> ### Author Response · Authors · 2024-11-25
> **Response to Reviewer S7t5 (part 1)**
>
> Thank you for your detailed feedback, which has helped strengthen our work. We have addressed each point you raised with specific clarifications and improvements.
>
> **1. The formulation in Sec.3 and Equation 1 are incomplete. \& The justification for modeling CoT as a joint probability distribution between different steps is not clear.**
>
> In the specified paragraph, $x \in D$ only represents a question or context, not include the responses. Modeling CoT as a joint probability is not new in the related literature[1,2].
>
> Specifically, we have update the related part that define clearly the notation of each step and claim that enhancing each middle step helps enhancing self alignment, referring to Line 160-183.
>
> [1] Training Chain-of-Thought via Latent-Variable Inference, NeurIPS 2023
>
> [2] Self-Consistency Improves Chain of Thought Reasoning in Language Models, ICLR 23
>
> **2. A dedicated subsection comparing AlignCoT with previous work is needed.**
>
> We have updated the related work section to clarify the differences between AlignCoT and existing self-alignment methods (Lines 83-95) and added a paragraph discussing how MoTE differs from fine-tuning approaches based on CoT (Lines 97-105).
>
> Our main observation is that existing self-alignment methods primarily focus on collecting optimal responses, while AlignCoT emphasizes that CoT is an effective approach for safety alignment, with the intermediate steps also playing a crucial role. Additionally, AlignCoT relies on supervised fine-tuning, avoiding the complexity of reinforcement learning (RL) techniques.
> Furthermore, we compare AlignCoT with modern self-alignment methods such as CoT, Critique-Revise, Mistake Analysis, RLCD, and MATRIX (see Table 1).
>
> Compared to methods that fine-tune with CoT data, MoTE is the first to leverage a Mixture of Experts (MoE) framework for CoT fine-tuning, introducing innovative techniques like shared experts and step-skipping to enhance alignment performance.
>
> **3. "Efficient training of MoTE" section offers little novelty.**
>
> We claim that applying masked attention in CoT training to enable step-skipping is a novel attempt. This approach is specifically designed to enhance the efficiency of MoTE training by allowing selective activation of experts for relevant steps, which distinguishes it from traditional masked attention techniques.
>
> **4. Clarity needs improvement.**
> > It is unclear which 7B LLM is referred to in Line 177 (Line 195 in updated manuscript). Is it Alpaca-7B?
>
> Yes.
>
> > The metrics used in Figure 2 are not specified or explained in the text.
>
> We have revised the manuscript and mentioned them in Line 197-200.
>
> Here we briefly introduce: the x-axis of Analysis Quality and Guidance Quality are scored by GPT-4-1106-preview. The x-axis of Single-step Inference and Multi-step Inference are the ratio of safe responses, following the matric **Harm** explained in Line 373.

---

> > ### Author Response · Authors · 2024-11-25
> > **Response to Reviewer S7t5 (part 2)**
> >
> > **5. Stronger Baseline \& other benchmark.**
> >
> > We have implemented MoTE on more advanced models, Llama2-7B[1] and Llama2-13B[2], and evaluated our results on the HH-RLHF and XSTest datasets.
> >
> > For HH-RLHF, we fine-tuned the two Llama2 models using MoTE with AlignCoT data and compared them with the current state-of-the-art, MATRIX. As shown in Table [X], our findings are as follows: (1) Llama2 models demonstrate superior safety alignment compared to other baselines; (2) MoTE achieves even better safety alignment when equipped with stronger LLMs on HH-RLHF.
> >
> > |                       | **Help** | **Harm** | **Harm-Help** |
> > |---------------------------|----------|----------|---------------|
> > | **current SoTA (MATRIX)** | 6.93     | 87.55    | 5.46          |
> > | **Llama2-7b**             | 6.98     | 89.01    | 6.41          |
> > | **MoTE-Llama2-7b**        | 7.11     | 93.27    | 6.94          |
> > | **Llama2-13b**            | 7.20     | 91.39    | 6.55          |
> > | **MoTE-Llama2-13b**       | 7.21     | 94.73    | 6.93          |
> >
> > **Table X**: Comparison of MoTE with Llama2-7B, Llama2-13B, and the Current State-of-the-Art on HH-RLHF.
> >
> > For XSTest, we report both safe refusal and unsafe refusal rates, following the methodology in Table 2 of their paper. As shown in Table [Y], MoTE consistently achieves lower refusal rates on safe prompts and higher refusal rates on unsafe prompts. This indicates a reduction in exaggerated safety behaviors. This result is particularly interesting and may be attributed to the "think-before-speak" step-by-step alignment approach, which significantly enhances helpfulness in handling harmful questions.
> >
> > | **Safe**            | **StrMatch$\downarrow$** | **GPT-4$\downarrow$** |
> > |---------------------|----------------------------------|-------------------------------|
> > | **Llama2-7b**       | 15.2                             | 28.4                          |
> > | **MoTE-Llama2-7b**  | 10.3                             | 25.8                          |
> > | **Llama2-13b**      | 2.8                              | 18                            |
> > | **MoTE-Llama2-13b** | 2                                | 15.6                          |
> > |              |                                  |                               |
> > | **Unsafe**          | **StrMatch$\uparrow$**     | **GPT-4$\uparrow$**     |
> > | **Llama2-7b**       | 40.5                             | 59.5                          |
> > | **MoTE-Llama2-7b**  | 60.7                             | 78.6                          |
> > | **Llama2-13b**      | 68.5                             | 78.2                          |
> > | **MoTE-Llama2-13b** | 81                               | 83.5                          |
> > **Table Y**: Comparison of MoTE with Llama2-7B, Llama2-13B on XSTest.
> >
> > [1] https://huggingface.co/meta-llama/Llama-2-7b-chat-hf
> >
> > [2] https://huggingface.co/meta-llama/Llama-2-13b-chat-hf
> >
> > **6. Code.**
> >
> > We will release the code upon acceptance.

---

> > > ### Comment · Reviewer_S7t5 · 2024-11-26
> > > **Thank you**
> > >
> > > Thank you for the efforts you spent on the rebuttal. I have decided to maintain my score. The submission can benefit from a major reversion based on the comments provided by all reviewers.

---

### Official Review · Reviewer_dHuL · 2024-11-05

**Soundness:** 2
**Presentation:** 2
**Contribution:** 2
**Rating:** 5
**Confidence:** 3

**Summary:**

This paper proposes a new AlignCoT dataset, which includes the original question, question analysis, answer guidance, and final safe answer by prompting the model with Chain of Thought (CoT). A new SFT method is also introduced, based on the AlignCoT dataset. This method incorporates multiple LoRA matrices corresponding to different parts: question, question analysis, answer guidance, and answer. Through comparative analysis, MoTE demonstrates better alignment efficacy compared to benchmark alignment techniques.

**Strengths:**

Strengths:

1. The innovative idea of using multiple LoRA experts.

2. Leveraging CoT and introducing AlignCoT.

**Weaknesses:**

Weaknesses:

1. The writing could be improved.

2. The MoTE method is heavily reliant on the AlignCoT dataset, making it difficult to extend to other datasets.

**Questions:**

I am unclear about the relationship between \(x\), \(x_a\), \(x_g\), \(y_{\text{cot}}\), and MoTE. From my understanding, this paper uses a CoT prompt to produce the question analysis first, followed by answer guidance, and finally the safe answer. Additionally, the paper uses this generated data as fine-tuning data for SFT on this model. Is the data generated by the model used to retrain the same model?

- Line 128: It is not clear whether the question analysis, answer guidance, and final safe answer are produced by the same model, or if question analysis and answer guidance can be generated by other models.

- Line 128: Where does the training data (\(x_a\), \(x_g\), \(y_{\text{cot}}\)) for SFT come from?

- Figure 2: Some subfigure titles have a period while others do not. Additionally, the x-axis is not clearly labeled.

- Lines 182-210: Should this part be located in the Experiments section? It seems to be an analysis of the method.

- Line 185: An introduction to the metrics would be helpful. In Table 1, there is no explanation for "Help," "Harm," and "Harm-Help."

- Line 367: What fine-tuning datasets are used for the baselines? \(D_a\), \(D_g\), \(D_{\text{ans}}\), etc.? If it is \(D_{\text{ans}}\), since MoTE is based on the AlignCoT dataset, this could lead to an unfair comparison.

---

> ### Author Response · Authors · 2024-11-25
> **Response to Reviewer dHuL**
>
> Thank you for your detailed feedback, which has helped strengthen our work. We have addressed each point you raised with specific clarifications and improvements.
>
> **1. Unclear about the relationship between (x), (x\_a), (x\_g), (y\_{\text{cot}}), and MoTE.**
>
> Your understanding is correct. We utilize the model itself to generate analyses, guidance, and safe responses for safety self-alignment. This process involves a three-step generation, which we treat as a structured thought chain designed to produce harmless responses effectively. The data generated through this process are then used to re-train the model, resulting in a safer model.
>
> MoTE is an efficient framework that assigns specialized expertise to each expert for the respective steps, enhancing the self-alignment process.
>
> **2. Line 128 (Line 139 in updated manuscript):  are analysis, guidance and safe answers are produced by the same model.**
>
> Yes. For self-alignment, we use self-generated data, including analysis, guidance, and safe answers, to refine and enhance the original model.
>
> **3. Line 128 (Line 139 in updated manuscript): where does training data ((x\_a), (x\_g), (y\_{\text{cot}})) come from?**
>
> All training data is generated by prompting the original model using AlignCoT, following the prompts outlined in Fig. 1. The harmful questions are sourced from the training sets of PKU-RLHF and HH-RLHF.
>
> **4. Figure 2: Subfigure Titles and X-axis.**
>
> We have updated the titles and metrics as described in Lines 197-200.
>
> The x-axis for Analysis Quality and Guidance Quality represents scores provided by GPT-4-1106-preview. The x-axis for Single-step Inference and Multi-step Inference shows the ratio of safe responses, measured using the metric \textbf{Harm} as explained in Line 359.
>
> **5. Line 182-210 (Line 203-236 in updated manuscript): move this part to the Experiment Section.**
>
> Thank you for the suggestion. We have moved Observation 1 to the Experiment section. However, we consider the other two observations crucial for motivating MoTE, so we have kept them in their original position.
>
> **6. Line 185 (Line 197 in updated manuscript) An introduction to the metrics would be helpful. In Table 1, there is no explanation for "Help," "Harm," and "Harm-Help."**
>
> We have update the manuscript to include an introduction to the metric in Line 197. For Table 1, All metrics are introduced in Line 369-377.
>
> Here we briefly introduce them:
> Helpfulness (Help) is rated on a score from 1 to 10 by GPT to determine the informativeness of responses. For harmlessness (Harm), a binary assessment by GPT determines the safety of answers, reporting a harmless rate. To ensure that higher harmlessness rates are not achieved by simply declining to answer, we also measure the helpfulness (Harm-Help) for responses to harmlessness prompts. Higher scores indicate better performance across all metrics.
>
> **7. Line 367 (Line 413 in the updated manuscript): what datasets are used for fine-tuning the baselines?**
>
> All baselines in Table 1 are fine-tuned using the datasets specified in their respective works. We highlight that existing safety alignment methods often overlook the importance of intermediate steps. By incorporating these steps through MoTE, we achieve superior performance, which constitutes our key contribution.
>
> Additionally, the result of training with D\_{\text{ans}} can be referenced in Figure 6 (AlignCoT w/o Analysis \& Guidance). The \textbf{Harm} score reaches approximately 80, approaching the performance of the state-of-the-art MATRIX.

---

### Meta-Review · Area_Chair_nvWk · 2024-12-19

**Metareview:**

**Summary**: This paper proposes MoTE (Mixture of insighTful Experts), a framework that combines the CoT reasoning method with a Mixture of Experts (MoE) architecture for improving safety alignment in LLMs. The authors introduce AlignCoT, a structured dataset that includes intermediate reasoning steps such as question analysis, answer guidance, and safe response generation, allowing fine-tuning through self-generated data. The MoTE framework builds on this by assigning specific LoRA experts to each step in the AlignCoT process, introducing shared experts and step-skipping for training efficiency.

**Strengths**:
- Using multiple LoRA experts is a novel idea
- Detailed ablation studies
- The paper is very-well organized and the presentation is clear

**Weaknesses**:
- Lack of clarity in the writing: Most reviewers asked for clarifications on the method and pointed out that the writing could be improved, to an extent where the paper would benefit from a heavy full-on revision. Moreover, Reviewer `S7t5` asked for extended coverage I the related work section in order to clarify the approach’s novelty compared to prior work.
- Limited novelty: Reviewers `S7t5` and `aK5g` complain about limited novelty — AlignCoT is just one of many CoT frameworks, novelty of MoTE compared to MoE is relatively limited, and moreover the benefit of the shared expert in MoTE is a bit unclear compared to MoE.
- Poor baseline comparison: Multiple reviewers asked for comparisons against newer more powerful models that better reflect the state-of-the-art in language model performance and alignment challenges, and are concerned about whether the results would hold in those cases.
- Limited reproducibility/generalization capabilities: Multiple reviewers pointed out ways in which the method may not be easy to replicate or generalize— reliance on the AlignCoT dataset, no code provided, narrow evaluation on PKU-SafeRLHF and HH-RLHF.

**Recommendation**:
Overall, the paper presented an interesting idea of using multiple LoRA experts and had detailed ablation studies, but unfortunately there are big concerns regarding writing clarify, novelty, inappropriate baseline comparison, and limited generalization. After the rebuttal, reviewers agree that the paper would benefit from a significant revision. As such, I vote to Reject this paper.

**Additional Comments On Reviewer Discussion:**

Despite the authors’ best efforts in the rebuttal, Reviewers `TQAf` and `S7t5` decided to maintain their rejection scores.

---

### Decision · Program_Chairs · 2025-01-22

Reject